# Aerosol Size Distribution Properties Associated with Cold-Air Outbreaks in the Norwegian Arctic

Abigail S. Williams[1], Jeramy L. Dedrick[1], Lynn M. Russell[1], Florian Tornow[2,3], Israel Silber[4,*], Ann M. Fridlind[3], Benjamin Swanson[5], Paul J. DeMott[5], Paul Zieger[6,7], and Radovan Krejci[6,7]

[1]Scripps Institution of Oceanography, University of California San Diego, La Jolla, CA, USA
[2]Center for Climate Systems Research, Columbia University, New York, NY, USA
[3]NASA Goddard Institute for Space Studies, New York, NY, USA
[4]Department of Meteorology and Atmospheric Science, Pennsylvania State University, University Park, PA, USA
[*]now at Atmospheric, Climate, and Earth Sciences Division, Pacific Northwest National Laboratory, Richland, WA, USA
[5]Department of Atmospheric Science, Colorado State University, Fort Collins, CO, USA
[6]Department of Environmental Science, Stockholm University, Stockholm, Sweden
[7]Bolin Centre for Climate Research, Stockholm University, Stockholm, Sweden

**Correspondence:** Lynn M. Russell (lmrussell@ucsd.edu)

**Abstract.** The aerosol particles serving as cloud condensation and ice nuclei contribute to key cloud processes associated with cold-air outbreak (CAO) events but are poorly constrained in climate models due to sparse observations. Here we retrieve aerosol number size distribution modes from measurements at Andenes, Norway, during the Cold-Air Outbreaks in the Marine Boundary Layer Experiment (COMBLE) and at Zeppelin Observatory, approximately 1000 km upwind from Andenes at Svalbard. During CAO events at Andenes, the sea spray mode number concentration is correlated to strong over-ocean winds with a mean of $8 \pm 4$ cm$^{-3}$ that is 71% higher than during non-CAO conditions. Additionally during CAO events at Andenes, the mean Hoppel minimum diameter is 6 nm smaller than during non-CAO conditions though the estimated supersaturation is lower and the mean number concentration of particles that likely activated in-cloud is $109 \pm 61$ cm$^{-3}$ with no statistically significant difference from the non-CAO mean of $99 \pm 66$ cm$^{-3}$. For CAO trajectories between Zeppelin Observatory and Andenes, the upwind-to-downwind change in number concentration is largest for the accumulation mode with a mean decrease of $93 \pm 95$ cm$^{-3}$, likely attributable primarily to precipitation scavenging. These characteristic properties of aerosol number size distributions during CAO events provide guidance for evaluating CAO aerosol-cloud interaction processes in models.

## 1   Introduction

The equatorward transport of polar air masses brings cold air over a relatively warmer open ocean, spurring large turbulent surface fluxes of heat and moisture and creating so-called marine cold-air outbreak (CAO) events. The forming clouds often appear first in the shape of rolls (Brümmer, 1999) before filling in towards a near-overcast deck and then transitioning into a more broken, open-cellular state further downwind (Abel et al., 2017; Lloyd et al., 2018; McCoy et al., 2017; Geerts et al., 2022). Such a cloud regime transition during CAO events is often seen in satellite imagery (e.g. Fig. 1 of Geerts et al. (2022)) and has important implications for the regional radiation budget (McCoy et al., 2017). Climate and numerical weather pre-

diction models struggle to accurately capture the clouds associated with CAO events. Such model limitations emerge since important dynamical aspects are unresolved (Field et al., 2017). In addition, there is evidence of strong cloud and aerosol microphysical processes that may modulate the macrophysical structure of clouds. For example, Tornow et al. (2021) found that mid-latitude CAO cloud transitions which resembled observations could only be simulated when the number concentration of aerosol serving as cloud condensation nuclei (CCN) was allowed to evolve with time, as aerosol can control the onset of transition-driving precipitation (Abel et al., 2017; Tornow et al., 2021). For polar CAOs of lower temperature, aerosol may additionally provide a reservoir for ice nuclei (e.g. Bigg (1996)). Previous modeling of cold-air outbreaks have typically relied on sparse observational constraints (e.g. Field et al. (2014); Abel et al. (2017); Tornow et al. (2023)), illustrating a need for observations of aerosol properties associated with CAO events in order to inform, constrain, and improve model simulations and forecasts.

Aerosol number concentrations in the Arctic are typically very low, with median concentrations consistently less than 400 $cm^{-3}$ (Croft et al., 2016; Pernov et al., 2022; Freud et al., 2017; Tunved et al., 2012). The annual maximum in number concentration at the Zeppelin Observatory in Svalbard is during spring due to an increase in accumulation mode particles (termed Arctic haze), with a maximum in Aitken mode particles during summer, and an annual minimum during fall (Croft et al., 2016; Pernov et al., 2022; Freud et al., 2017; Tunved et al., 2012; Schmale et al., 2022). Aerosols in this region typically have chemical compositions that largely consist of sulfates, sea salt, and organic components (Willis et al., 2018; Moore et al., 2011; Adachi et al., 2022; Schmale et al., 2022; Moschos et al., 2022). The variability of aerosol properties is often associated with differences in the meteorological conditions and transport patterns the aerosol experienced, referred to as the air mass history (Schmale et al., 2022; Pernov et al., 2022; Freud et al., 2017; Tunved et al., 2012). Since CAO events generally have air mass histories that are similar to each other, we could expect that their aerosol properties represent only a subset of properties observed in Arctic conditions. Previous studies have reported the accumulation mode number concentration of particles as approximately 100 $cm^{-3}$ upwind and approximately 25 $cm^{-3}$ downwind of the transition in cloud structure during CAO events (Abel et al., 2017; Lloyd et al., 2018; Sanchez et al., 2022). These results, however, are limited to only four case studies and provide no information on the aerosol number size distribution associated with CAO events.

CCN consist primarily of accumulation mode particles, with smaller particles serving as CCN when accumulation mode number concentrations are very low or supersaturations are high (Jung et al., 2018; Karlsson et al., 2022; Koike et al., 2019; Bulatovic et al., 2021; Zábori et al., 2015). Known as cloud-processing, an addition of aerosol mass can result from the aqueous production of a component from the vapor phase, such as sulfate from $SO_2$ (Isokääntä et al., 2022), resulting in particles that are larger after the cloud has evaporated than before it formed (Freud et al., 2017; Croft et al., 2016; Zheng et al., 2018). Precipitation acts as a primary sink for accumulation mode particles (Isokääntä et al., 2022; Croft et al., 2016; Zheng et al., 2018; Freud et al., 2017; Tunved et al., 2012), and this scavenging has been observed in association with CAO events, although limited to three case studies and aerosols of diameters between 0.1 and 3.0 $\mu$m (Abel et al., 2017; Lloyd et al., 2018). In mid-latitude clouds off the Eastern seaboard, dilution of accumulation mode particles by rapid boundary layer growth has also been shown to contribute to lowering droplet number concentrations with increasing fetch off-shore (Tornow et al., 2022).

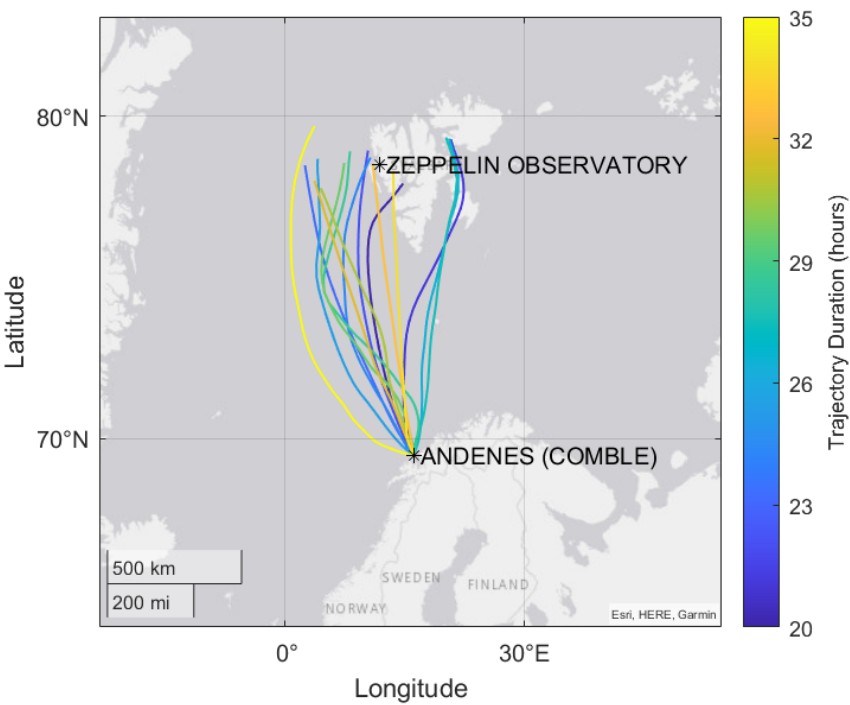

**Figure 1.** Map illustrating the locations (black asterisks) of Andenes in Norway and the Zeppelin Observatory in Svalbard. Overlaid are 15 trajectories that occur during 8 CAO events at Andenes (Table S1). Trajectories are colored by the amount of time between closest passage by Zeppelin Observatory and arrival at Andenes.

In order to better understand the interactions between aerosols and CAO events, we have studied aerosol number size distributions measured in Andenes, Norway (69.30°N, 16.15°E) during the Cold-Air Outbreaks in the Marine Boundary Layer Experiment (COMBLE; Geerts et al. (2022)). We additionally examined aerosol measurements from the Zeppelin Observatory in Svalbard (78.90°N, 11.88°E), located approximately 1000 km upwind of Andenes (Fig. 1), during time periods when CAO air mass trajectories passed near the Zeppelin Observatory before arriving at Andenes. This allowed for an evaluation of the consistencies and variability of the cloud-processed aerosol number size distribution during CAO events versus non-CAO conditions at Andenes and between upwind (Zeppelin Observatory) and downwind (Andenes) conditions for certain CAO events. We investigate the aerosol properties associated with CAO events, including the sea spray aerosol concentration (Sect. 3), the fraction of particles that can activate to cloud droplets (Sect. 4), and the variation of modal number concentration with precipitation (Sect. 5).

## 2 Methods

Section 2.1 describes the selection of CAO air masses that passed in close proximity to Zeppelin Observatory in Svalbard and traversed $\sim 1000$ km downwind to Andenes, Norway. Section 2.2 discusses the retrieval of the aerosol mode properties associated with the measured size distributions at Andenes and Zeppelin Observatory. Lastly, Section 2.3 describes the measurements of the trace gases and meteorological properties used in this work.

### 2.1 CAO events

As identified in Geerts et al. (2022), 49 CAO events occur at Andenes during COMBLE between 01 December 2019 and 31 May 2020. The 49 CAO events account for 18.6% of the entire COMBLE duration, and the remaining 81.4% is referred to as non-CAO conditions. Sections 3 and 4 utilize these CAO and non-CAO time periods. 120-hour back-trajectories were calculated every 3 hours between 01 December 2019 and 31 May 2020, initiated at Andenes (surface level; 2 meters above mean sea level; 69.30°N, 16.15°E), using the Hybrid Single-Particle Lagrangian Integrated Trajectory (HYSPLIT; Stein et al. (2015)) with the fifth generation European Centre for Medium-Range Weather Forecasts (ECMWF) atmospheric reanalysis (ERA5; Hersbach et al. (2020)) at 1 degree and 1-hour spatial and temporal resolutions. An ensemble is run for each trajectory, in which each member of the ensemble is calculated by offsetting the meteorological data by 7.5 km in the horizontal directions (a total of nine ensemble members).

23 (47%) of the CAO events during COMBLE have trajectories that pass within 200 km of Zeppelin Observatory in Svalbard (78.90°N, 11.88°E). Section 5 focuses on 8 (16%) of the CAO events during COMBLE that have complete aerosol number size distribution measurements available at both Zeppelin Observatory and Andenes. These 8 CAO events include 15 of the calculated 3-hourly back-trajectories shown in Fig. 1. Since a few of these 15 selected back-trajectories remain at surface level between Andenes and Zeppelin Observatory, all trajectories were initiated at multiple vertical levels in 50 m increments above the surface. For the 15 selected back-trajectories, all still pass within 200 km of Zeppelin Observatory when initiated at a height above the surface that never intersects the surface at any point between Andenes and Zeppelin Observatory (Fig. S1). Moreover, the trajectories identified as not meeting the criteria of passing within 200 km of Zeppelin Observatory are unchanged for surface or higher initial heights.

### 2.2 In-situ aerosol measurements

Aerosol properties were measured at Andenes during COMBLE by the Aerosol Observing System (AOS) operated by the U.S. Department of Energy (DOE) Atmospheric Radiation Measurement (ARM) Facility. More details on the general set-up of the AOS can be found in Uin et al. (2019). Particle number size distributions were provided by an Ultra High Sensitivity Aerosol Spectrometer (UHSAS; Droplet Measurement Technologies Inc.; Uin (2016a)) and a Scanning Mobility Particle Sizer (SMPS; TSI Inc., Model 3936; Kuang (2016a)). The UHSAS utilizes optical-scattering to size and count particles between 60 and 1000 nm every 10 seconds while the SMPS uses electrical mobility to size particles between 10 and 500 nm every 5 minutes. The particle sizing uncertainty for both the UHSAS and SMPS is within $\pm$ 2.5% (Uin, 2016a; Kuang, 2016a). A condensation

particle counter (CPC; TSI Inc., Model 3772; (Kuang, 2016b)) recorded the total number concentration of submicron particles greater than 10 nm in diameter. The sample flows of the UHSAS, SMPS, and CPC were dried using Nafion dryers (Uin et al., 2019), and the aerosol inlet had a particle transmission efficiency of 100% between 10 and 1000 nm (Bullard et al., 2017).

At Zeppelin Observatory, particle number size distributions in the mobility diameter range of 5 - 809 nm were collected approximately every 15 min by a custom closed-loop differential mobility particle sizer (Platt et al., 2022; Karlsson et al., 2021; Tunved et al., 2012). An optical particle size spectrometer (OPSS, FIDAS 200 S, Palas GmbH, Germany) measured the size distribution of larger particles in the optical diameter range of 0.2 - 18 μm every 3 min under dry conditions. The COMBLE campaign coincided with the Ny-Ålesund Aerosol Cloud Experiment 2019-2020 (NASCENT) campaign at Ny-Ålesund, Svalbard (Pasquier et al., 2022). More details on the long-term measurements and the general set-up at Zeppelin Observatory can be found in Platt et al. (2022).

Aerosol mass was collected at Zeppelin Observatory using a three-stage filter pack (with a teflon Zefluor 2 μm pore, 47mm diameter filter and a cellulose Whatman 40 filter) every 24 hr (Lee et al., 2020; Platt et al., 2022). Due to the absence of a cyclone or impactor in front of the filter pack, there was no calibrated upper size cutoff (Lee et al., 2020; Ahn et al., 2021), although contributions from supermicron particles were expected to be small at this location. Following collection, filters were analyzed in the laboratory using ion chromatography to determine the mass of $SO_4^{2-}$, $NO_3^-$, $Cl^-$, $NH_4^+$, $Ca^{2+}$, $Mg^{2+}$, $K^+$, and $Na^+$ (Platt et al., 2022; Lee et al., 2020; Ahn et al., 2021). The mass concentration of sea salt was calculated as $1.47[Na^+]$ + $[Cl^-]$ as in Lewis et al. (2023) and Frossard et al. (2014).

During COMBLE at Andenes, a Humidified Tandem Differential Mobility Analyzer (HTDMA; Brechtel Manufacturing Inc. Model 3002; Uin (2016b)) recorded size distributions of humidified (RH = 80%) particles at five different initial dry (RH < 2%) sizes at 10 min intervals, from which the growth factors (GF; ratio of wet to dry diameter) for each of the five initial dry diameters were calculated. The HTDMA was calibrated at the beginning and end of the six-month COMBLE campaign by ARM instrument mentors, including verifying flow rates, particle sizing, and GF values of ammonium nitrate or ammonium sulfate particles against Köhler theoretical values (Uin, 2016b; Uin et al., 2019). The hygroscopicity parameter ($\kappa$) was calculated using the median GF of each distribution for the five initial dry diameters using Equation 11 of Petters and Kreidenweis (2007). Taking into account that the HTDMA has a particle sizing uncertainty of $\pm$ 1% and a RH uncertainty of $\pm$ 4% (Uin, 2016b), a Monte Carlo approach provided an estimate of the associated uncertainty in our calculated $\kappa$ values, with an approach similar to that of Schulze et al. (2020) and Massling et al. (2023). For each HTDMA measurement, 1000 iterations of $\kappa$ calculations were performed using input parameter values that were randomly selected from a normal distribution centered at the measured input parameter value with a standard deviation of its instrument-based uncertainty. The uncertainty in $\kappa$ is $\pm$ 24%, which is the standard deviation of the $\kappa$ distribution that results from the 1000 iterations.

During COMBLE at Andenes, a cloud condensation nuclei (CCN) counter (Droplet Measurement Technologies Model CCN-200; Uin (2016c)) provided measurements of CCN number concentration at 0.4% supersaturation every 15 minutes throughout the campaign. The CCN counter was was calibrated at the beginning and end of the six-month COMBLE campaign by ARM instrument mentors, including applying calibration coefficients determined from counting size-selected ammonium

sulfate particles before and after activation at various instrument temperatures (Uin, 2016c). The associated particle sizing uncertainty for the CCN counter is $\pm$ 0.25 μm (Uin, 2016c).

    Particle number size distribution measurements were averaged to 1- and 2-hour intervals and merged from the UHSAS and SMPS during COMBLE and the DMPS and OPSS at Zeppelin Observatory following the methods of Dedrick et al. (2022a), Modini et al. (2015) and Khlystov et al. (2004). There is a strong correlation between the measured CN concentration and total

particle number concentration integrated from the merged particle number size distributions at Andenes (r=0.80, Fig. S2) and at Zeppelin Observatory (r=0.95, Fig. S2).

    Three lognormal modes were fit to the merged number size distributions at both sites using an automated algorithm that calculates the modal number concentration (N), geometric mean diameter ($D_g$), and geometric standard deviation ($\sigma_g$) as parameters of the fitted distribution (Dedrick et al., 2022a; Modini et al., 2015; Saliba et al., 2019). The fitting algorithm, as

described in Dedrick et al. (2022a), is informed and constrained by supermicron scattering measurements taken by a three-wavelength integrating nephelometer during COMBLE (Uin, 2016d). A chi-square test was utilized to evaluate the quality of the calculated mode fits to the merged number size distributions, similar to Dedrick et al. (2022a). Approximately 20% of fits had a chi-square value that exceed the critical value at the 95% confidence interval, and these fits were manually adjusted by visual inspection or excluded.

The choice of three lognormal modes has also been used in prior studies in this region (e.g. Freud et al. (2017); Tunved et al. (2012)). We designate the three modes as the Aitken mode ($0.02\mu m < D_g < 0.1\mu m$), accumulation mode ($0.1\ \mu m < D_g < 0.3\mu m$), and sea spray mode ($0.3\mu m < D_g < 0.8\mu m$). This sea spray mode centered at diameters between 0.3 and 0.8 μm is consistent with previous reports of sea spray mode mean diameters ranging between 0.14 and 0.6 μm in field measurements (Russell et al., 2023; Modini et al., 2015; Dedrick et al., 2022a), and often appears as a "shoulder" in the distribution (Zheng

et al., 2018; Dedrick et al., 2022a; Modini et al., 2015). The sea spray mode mass is calculated from the fitted sea spray mode number assuming spherical particle homogeneity and a constant sea spray density of 2.0 g cm⁻³ for dry conditions (Zieger et al., 2017). The Pearson correlation coefficient, r, indicates a strong correlation (r = 0.92) of the fitted sea spray mode mass with the measured sea salt mass concentration at Zeppelin Observatory (Fig. 2). There is also a strong correlation between the fitted sea spray mode mass and measured sodium ion mass concentration (r = 0.90), and between the fitted sea spray mode

mass and measured chloride ion mass concentration (r = 0.94).

    The Hoppel minimum diameter ($D_{HM}$; local minimum in concentration located between the peaks of the Aitken and accumulation modes, Hoppel et al. (1986)) is retrieved for all measured aerosol particle number size distributions at Andenes following the methods presented by Dedrick et al. (2024). Using the $D_g$ of the fitted Aitken and accumulation modes as upper and lower diameter bounds, $D_{HM}$ is taken as the diameter bin of the merged measured number size distribution that has the lowest particle

number concentration within these diameter bounds. The instrument-based sizing uncertainty of $\pm$ 2.5% is estimated for the uncertainty in retrieved $D_{HM}$ values.

    The effective supersaturation for cloud processing is estimated using measurements of $D_{HM}$ and $\kappa$ at Andenes in Equation 10 of Petters and Kreidenweis (2007). $\kappa$ is interpolated to the diameter range of $D_{HM}$ using the HTDMA-derived $\kappa$ measurements at the five initial dry particle diameters of 50, 100, 150, 200, and 250 nm. Due to this limited size range of $\kappa$, effective

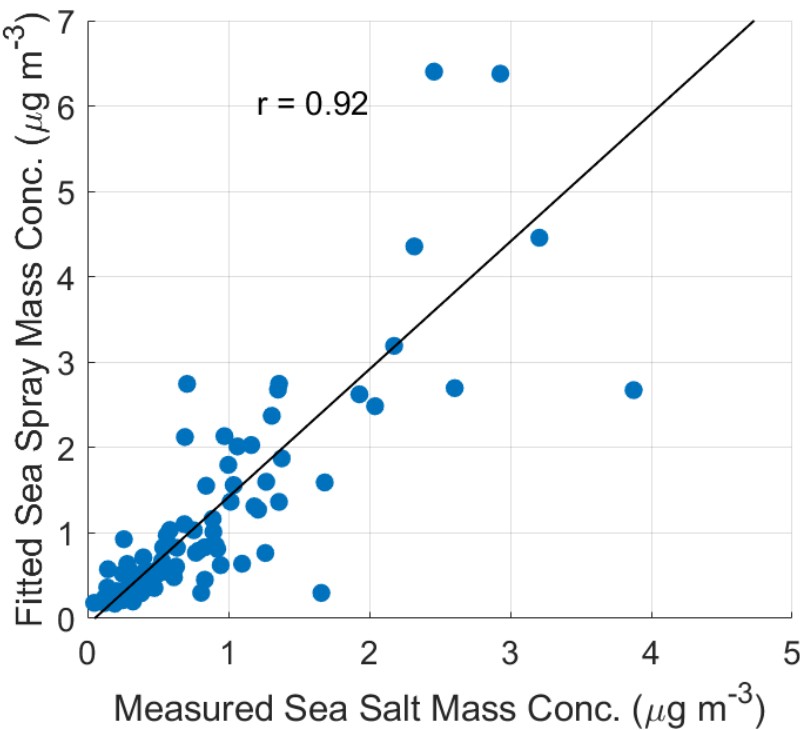

**Figure 2.** Scatter plot of the mass concentration of the fitted sea spray mode against the measured sea salt mass concentration at Zeppelin Observatory. The linear regression line is shown in solid black.

supersaturation is only calculated for points where 50 nm $<$ $D_{HM}$ $<$ 250 nm. Given the uncertainties in $D_{HM}$ ($\pm2.5\%$) and $\kappa$ ($\pm24\%$), a Monte Carlo approach (described above for the $\kappa$ uncertainty calculation) is used to estimate the uncertainty in the effective supersaturation to give an uncertainty of less than $\pm15\%$ for the calculated effective supersaturation values.

### 2.3 Ancillary variables

Carbon monoxide (CO) concentrations were measured at Andenes during COMBLE with a Los Gatos Research CO analyzer at
1-second resolution (Springston, 2015), and at Zeppelin Observatory with a Picarro G2401 cavity ring-down spectroscope at 1-minute resolution (Platt et al., 2022). Wind direction and speed were measured every 1 second by a Vaisala weather transmitter (WXT 520) mounted on the aerosol inlet at Andenes during COMBLE (Kyrouac, 2019). For direct comparison with aerosol measurements, wind and CO measurements were averaged to 1- and 2-hour intervals.

Mean surface precipitation rates with a resolution of 4 min at Andenes were obtained from the *precip_mean* variable of
the KAZRARSCL (Active Remote Sensing of CLouds Product Using Ka-band ARM Zenith Radars; Johnson et al. (2023)) value-added product (VAP) produced by the Department of Energy (DOE) Atmospheric Radiation Measurement User Facility (ARM). This variable is informed by measurements of the tipping bucket rain gauge that is part of the surface meteorological

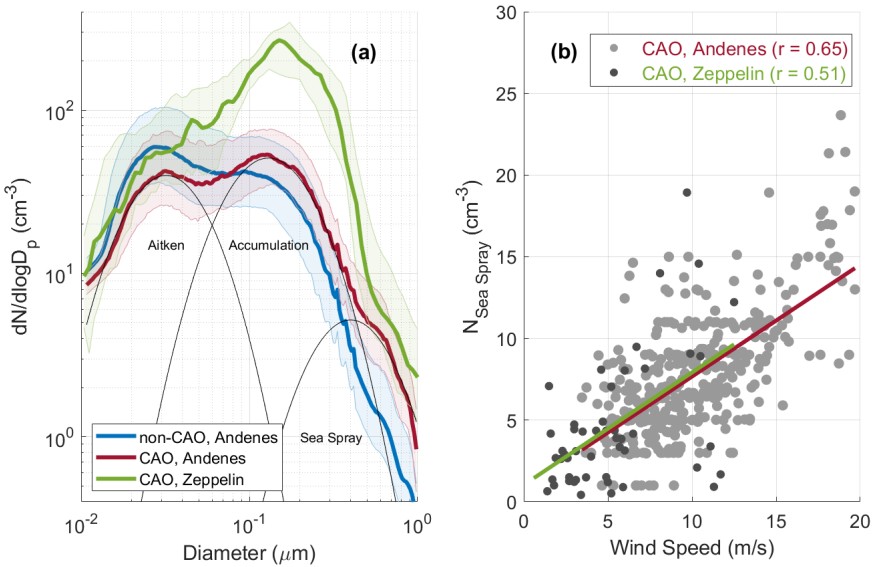

**Figure 3.** (a) Median particle number size distributions (thick solid lines) with 25th and 75th percentiles (shaded regions) for non-CAO conditions at Andenes (blue), CAO events at Andenes (red), and CAO event backtrajectories at Zeppelin Observatory (green). The median CAO distribution at Andenes is overlaid with the Aitken, accumulation, and sea spray modes (thin black lines). (b) Scatter plots of sea spray mode number concentration against measured wind speed. Points are 2-hr means during CAO events at Andenes (light gray points) or when CAO event backtrajectories pass near Zeppelin Observatory (dark gray points) and are overlaid with the corresponding regression line for Andenes and Zeppelin Observatory (red and green lines, respectively).

instrumentation (MET; Ritsche (2011)). Based on the mean precipitation rate, the amount of precipitation (the liquid-equivalent of all hydrometeors) accumulated at Andenes within the previous 24-hours at 1- and 2-hour intervals were calculated during

COMBLE.

## 3   Greater sea spray particle number concentration

The median aerosol particle number size distribution at Andenes is described by three lognormal modes (Aitken, accumulation, and sea spray) centered at approximately 0.035, 0.14, and 0.45 $\mu$m (Fig. 3a). The mean Aitken mode number concentration is $52 \pm 54$ cm$^{-3}$ during CAO events, which is 14% lower than during non-CAO conditions (Table 1). Conversely, the mean

accumulation mode number concentration is $109 \pm 75$ cm$^{-3}$, with 10% higher concentrations during CAO events compared to non-CAO conditions. These differences, however, are not statistically significant (hereafter statistically significant implies $p<0.05$ by two-sample t-test) and suggest that the number concentration of the two modes are controlled by independent factors rather than the aerosol co-varying with CAO events.

In contrast, the sea spray mode has a statistically-significant 71% higher mean number concentration of $8 \pm 4$ cm$^{-3}$ during

CAO events compared to non-CAO conditions (Table 1). This is likely associated with stronger over-ocean winds that are

**Table 1.** Summary of non-CAO versus CAO values for modal number concentration (N), wind direction, and wind speed. The modal number concentration values are the mean±SD. The wind direction values are the range of measurements, and the wind speed values are the percentage of time that hourly mean wind speed exceeds 10 m/s. Additionally reported for each variable is if the non-CAO and CAO values have a statistically-significant difference by t-test.

| Variable | non-CAO | CAO | Statistically-significant difference? |
|---|---|---|---|
| Aitken N (cm$^{-3}$) | $61 \pm 74$ | $52 \pm 54$ | No |
| Accumulation N (cm$^{-3}$) | $98 \pm 80$ | $109 \pm 75$ | No |
| Sea spray N (cm$^{-3}$) | $5 \pm 4$ | $8 \pm 4$ | Yes |
| Wind direction ($^{o}$) | 100-230 (southeasterly) | 250-30 (northwesterly) | Yes |
| Wind speed (% time >10 m/s) | 7 | 45 | Yes |

typical of CAO events (Kolstad, 2017). During COMBLE, CAO trajectories generally have spent the most recent 15-40 hours over the open ocean and arrive at Andenes with sustained winds from the northwest ($250^{o}$-$30^{o}$ defined in Geerts et al. (2022); Fig. 4b,d). Such CAO winds are from nearly the opposite direction of winds typical during non-CAO conditions, which come from over Scandinavia to the south and southeast ($100^{o}$-$230^{o}$) approximately 70% of the time (Fig. 4c). Binning the sea

spray mode number concentration by wind direction ($w_{dir}$) as either marine ($250^{o}>w_{dir}>30^{o}$, northwesterly) or continental ($100^{o}>w_{dir}>230^{o}$, southeasterly) reveals a mean number concentration of $8 \pm 4$ cm$^{-3}$ for marine wind directions and $5 \pm 2$ cm$^{-3}$ for continental wind directions. This statistically significant difference associated with wind direction suggests that the observed 71% increase in sea spray mode number concentration during CAO events is associated with the north-to-westerly direction of the winds.

More specifically, the higher sea spray mode number concentration is likely related to the wind speed, as sea spray mode mass concentration has previously been correlated to wind speed (Monahan, 1968; Russell et al., 2023). Hourly mean wind speed exceeds 10 ms$^{-1}$ for more than 45% of the time during CAO events, compared to only 7% during non-CAO periods (Fig. 4c-d). For CAO events, sea spray number concentration is positively correlated to local wind speed at Andenes (Pearson correlation coefficient (r = 0.65, p<0.05) and at Zeppelin Observatory (r=0.51, p<0.05; Fig. 3b). The mass concentration of the

sea spray mode is also correlated to wind speed at Andenes and at Zeppelin Observatory (r=0.62 and r=0.50, respectively; not shown). For non-CAO conditions at Andenes, the correlations between sea spray number concentration and wind speed and between sea spray mass concentration and wind speed are weaker than during CAO events (r = 0.38 and r = 0.24, respectively; not shown). These relationships show that the higher sea spray mode concentration during CAO events co-varies with the higher over-ocean wind speed associated with CAO events.

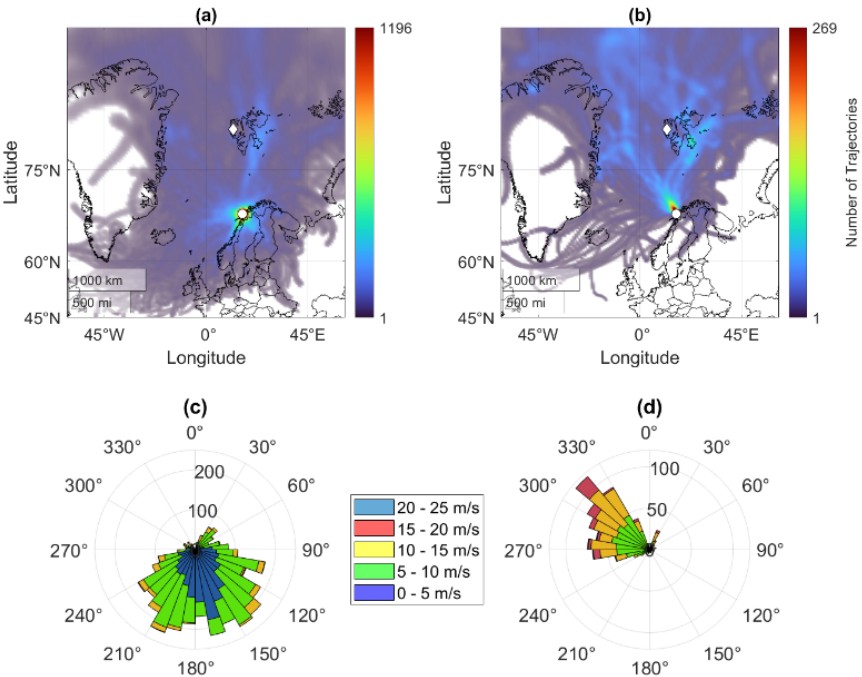

**Figure 4.** (a-b) Density plots of backtrajectories in 1 degree latitude and longitude bins originating at Andenes, Norway during (a) non-CAO conditions and (b) CAO events. Andenes is marked with a white circle, Zeppelin Observatory is marked with a white diamond. (c-d) Rose histogram plots of wind direction in 10° bins with stacked 5 m/s bins of wind speed during (c) non-CAO conditions and (d) CAO events.

## 4  Activation of particles to cloud drops

In regions where the aerosol particle number size distribution has been processed by low clouds, the Hoppel minimum diameter ($D_{HM}$) provides an indicator of the size of the smallest particles that were activated to and grown by processing as cloud droplets (Hoppel et al. (1986); Krüger et al. (2014); this interpretation of $D_{HM}$ as the minimum diameter activated does not account for cloud processing that is too short to result in particle growth from the Aitken mode). Median $D_{HM}$ during non-CAO conditions is 59 nm (Fig. 5), which falls within the reported range of approximately 50-80 nm for Arctic and marine-influenced aerosol number size distributions (Hoppel et al., 1986; Pernov et al., 2022; Freud et al., 2017; Tunved et al., 2012). During CAO events, the median $D_{HM}$ is 53 nm, which is 6 nm smaller than during non-CAO conditions. This difference is statistically significant and may be attributed to differences in supersaturation driven by meteorological or aerosol microphysical properties (Reutter et al. (2009); Ghan et al. (1998); Chen et al. (2016); Text S1).

The median hygroscopicity parameter ($\kappa$) at particle diameters between 100 and 250 nm is $0.11 \pm 0.02$ higher during CAO events compared to non-CAO conditions (Fig. 5). At 50 nm, however, the median $\kappa$ is 0.2 higher during CAO events with a value of 0.4. A diameter of 50 nm falls at the 43[rd] percentile of $D_{HM}$ during CAO events and at the 31[st] percentile of $D_{HM}$ during non-CAO conditions, implying that more particles at 50 nm may be cloud-processed during CAO events. The enhanced

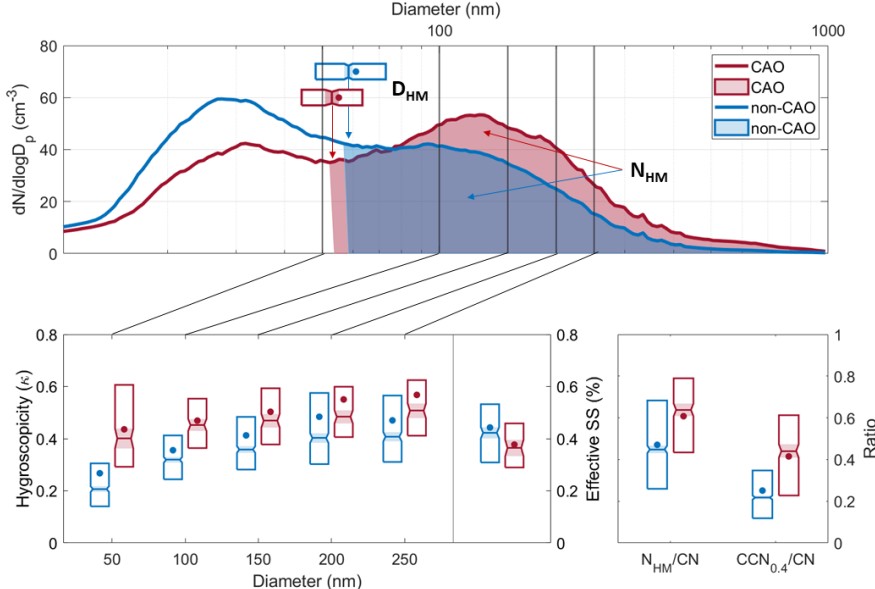

**Figure 5.** (top) The median number size distributions for non-CAO conditions (blue) and CAO events (red) with shaded regions indicating the integrated number concentration above the Hoppel minimum ($N_{HM}$), with horizontal boxplots of Hoppel minimum diameter, $D_{HM}$. Overlaid are (bottom left) boxplots of HTDMA-derived kappa at dry particle diameters of 50, 100, 150, 200, 250 nm and effective supersaturation calculated from $D_{HM}$ and HTDMA-derived kappa and (bottom right) boxplots of the ratio of $N_{HM}$ to CN and $CCN_{0.4}$ to CN. For all boxplots, notched center line is the median, lower and upper edges extend to the 75th and 25th percentiles, and circle marker is the mean.

hygroscopicity at this diameter could indicate a contribution from sulfate ($\kappa = 0.84$, Sanchez et al. (2018)) added to activated
particles during cloud processing (Crumeyrolle et al., 2008) or from differences in particle sources and transport patterns.

The diameter of the smallest particle activated is determined by the highest critical supersaturation for a given aerosol composition, making $D_{HM}$ an indicator of the effective supersaturation in clouds that persisted long enough for processing to cause growth. If the aerosol chemical composition and by consequence its hygroscopicity, represented by $\kappa$, were the same between CAO events and non-CAO conditions, then the smaller $D_{HM}$ observed during CAO events would imply a higher effective
supersaturation during CAO events (Fig. 6). However, if $D_{HM}$ were the same between CAO events and non-CAO conditions, then the higher $\kappa$ observed during CAO events would imply a lower effective supersaturation during CAO events (Fig. 6). There is a statistically-significant lower median effective supersaturation of 0.37% during CAO events compared to a median of 0.41% during non-CAO conditions (Fig. 5). Interestingly, this suggests that for the range of $D_{HM}$ and $\kappa$ observed during the CAO events that the higher hygroscopicity ($\kappa$) has a larger effect on supersaturation than does the smaller particle size ($D_{HM}$),
resulting in the observed lower effective supersaturation during CAO events compared to during non-CAO conditions (Text S2).

The mean number of particles that were activated and processed by clouds, represented by the integrated number concentration above $D_{HM}$ (designated as $N_{HM}$), is $109 \pm 61$ cm$^{-3}$ during CAO events with no statistically significant difference from

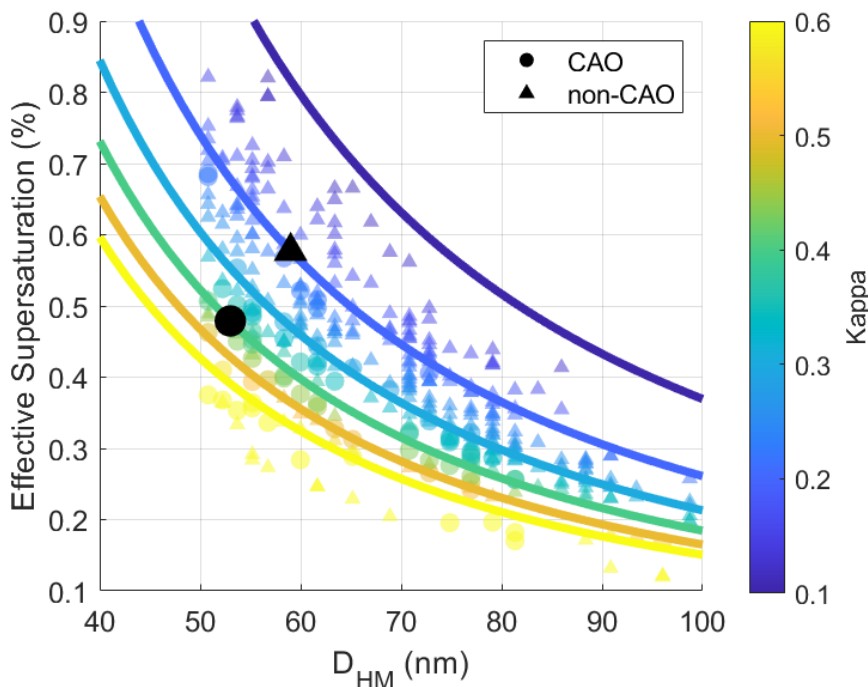

**Figure 6.** Scatter plot of effective supersaturation against $D_{HM}$ colored by hygroscopicity ($\kappa$). Measurements during CAO events are represented by circles and during non-CAO conditions are represented by triangles. Overlaid are lines of constant hygroscopicity ($\kappa$) at intervals of 0.1 between 0.1 and 0.6, as described by the relationship amongst supersaturation, diameter, and $\kappa$ given in Petters and Kreidenweis (2007). In black are the points corresponding to the median $D_{HM}$ and median hygroscopicity ($\kappa$) at 50 nm for CAO events and non-CAO conditions. Measurements for $D_{HM} < 50$ nm are omitted since there are no HTDMA-derived $\kappa$ measurements below this size.

its mean of $99 \pm 66$ cm$^{-3}$ during non-CAO conditions. There is a strong and significant positive correlation (r = 0.83; p < 0.05)
between $N_{HM}$ and CCN at 0.4% supersaturation ($CCN_{0.4}$) during COMBLE (Fig. 7a). This is in agreement with the calculated effective supersaturation mean of 0.42% during COMBLE. The effective supersaturation is correlated (r = 0.6) to $N_{HM}$ -$CCN_{0.4}$, where positive values of $N_{HM} - CCN_{0.4}$ represent effective supersaturations above 0.4% and negative values represent effective supersaturations below 0.4% (Fig. 7b). The mean $CCN_{0.4}$ is $90 \pm 50$ cm$^{-3}$ during CAO events, which is within the reported range of approximately 35-176 cm$^{-3}$ for mean CCN concentrations measured in the Arctic at 0.3-0.5% supersaturation
(Table S2; Jung et al. (2018); Paramonov et al. (2015); Moore et al. (2011); Dall´osto et al.; Herenz et al. (2018); Lathem et al. (2013); Zábori et al. (2015); Martin et al. (2011); Massling et al. (2023); Duplessis et al. (2024)).

CAO events are associated with cleaner condensation nuclei (CN) concentrations with a mean of $218 \pm 134$ cm$^{-3}$, which is lower with statistical significance than the mean of $311 \pm 234$ cm$^{-3}$ during non-CAO conditions. As a consequence of similar $N_{HM}$ concentrations between CAO events and non-CAO conditions and lower CN concentrations during CAO events, the mean
ratio $N_{HM}/CN$ is $0.61 \pm 0.25$ during CAO events, which is greater with statistical significance than the mean of $0.47 \pm 0.26$

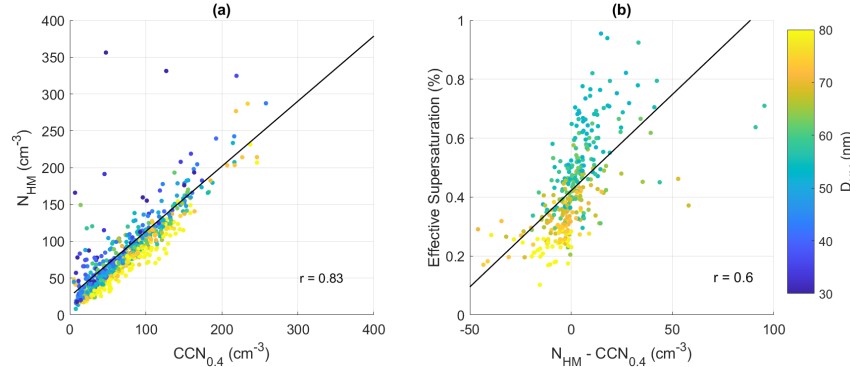

**Figure 7.** (a) Scatter plot of $N_{HM}$ against $CCN_{0.4}$. (b) Scatter plot of effective supersaturation (calculated with $D_{HM}$ and HTDMA-derived kappa for points with $D_{HM} > 50$ nm) against the difference between $N_{HM}$ and CCN at 0.4% supersaturation and colored by $D_{HM}$. For both plots, the corresponding linear regression lines are shown in solid black.

during non-CAO conditions (Fig. 5). Given the assumption that all $N_{HM}$ are activated to cloud droplets, an approximately 15% higher fraction of ambient CN activate during CAO conditions. The range of $N_{HM}$/CN shown in Figure 5 falls within reported ratios of $N_{CCN}/N_{CN}$, which varied between approximately 0.1 and 0.9 for CCN at 0.3-0.5% supersaturation at high-latitudes (Jung et al., 2018; Moore et al., 2011; Paramonov et al., 2015; Zábori et al., 2015; Lathem et al., 2013).

## 5 Scavenging by precipitation

To characterize aerosol changes along CAO trajectories, we compared pairs of particle number size distributions measured upwind at Zeppelin Observatory and downwind at Andenes for 15 CAO trajectories that passed within 200 km of Zeppelin Observatory during eight CAO events at Andenes (Fig. 1; Table S1). The mean and standard deviation of the change in the aerosol particle number size distribution from upwind to downwind sites is shown in Figure 8. The largest change in $dN/dlogD_p$ occurs within the accumulation mode size range, with a mean decrease of 203 cm$^{-3}$ at a diameter of 142 nm. The number concentrations retrieved from the fitted accumulation modes of the upwind and downwind size distributions are used to show a mean accumulation mode total number concentration decrease of $93 \pm 95$ cm$^{-3}$. This change is larger than previous reports of total aerosol number concentration decreases observed in three CAO cases (Abel et al., 2017; Lloyd et al., 2018; Sanchez et al., 2022).

We attribute the majority of the observed decrease in accumulation mode particles during the CAO events to precipitation scavenging, since carbon monoxide (CO) concentration along trajectories decreased by only 7.3% at the 75[th] percentile compared to an accumulation number decrease of 71.8% at the 75[th] percentile. The larger decrease in accumulation mode particles than in CO concentrations supports the attribution of losses in aerosol number to precipitation since CO is typically conserved during precipitation (e.g.Garrett et al. (2010); Dadashazar et al. (2021)). However, boundary layer growth in CAO cases can introduce uncertainty in this accounting. Of the 15 CAO trajectories identified here, one case shows an increase in accumula-

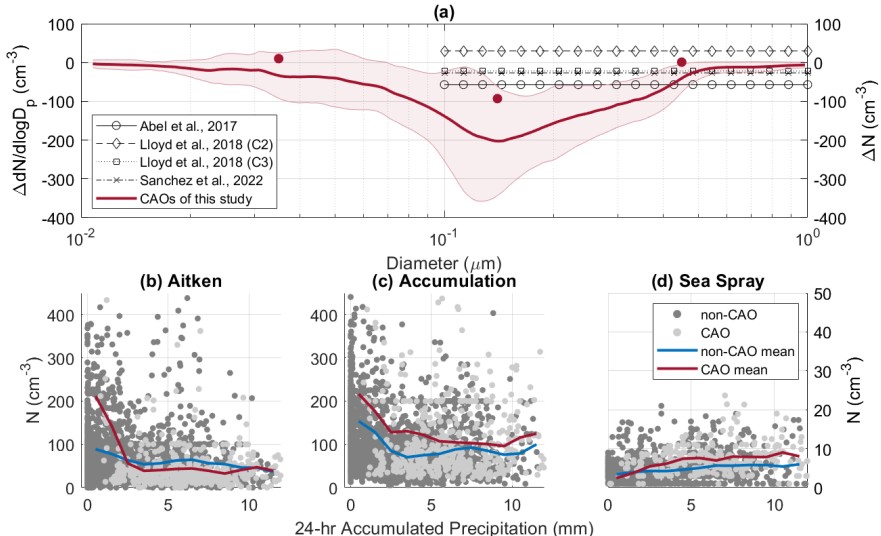

**Figure 8.** (a) Left axis: The red solid line represents the mean and the red shaded area represents the standard deviation of the change in the aerosol particle number size distribution from the upwind Zeppelin Observatory site to the downwind Andenes site for the 15 CAO trajectories of this study (Fig. S3, Table S1). Right axis: Red dots represent the mean change in total aerosol number concentration for each fitted mode, the Aitken, accumulation, and sea spray modes from left to right, for the 15 CAO trajectories of this study. Overlaid in black lines with markers are the mean change in total aerosol number concentration reported for the case of Abel et al. (2017), cases two and three of Lloyd et al. (2018) and the closed- to open-cell case of Sanchez et al. (2022). These studies measured the total number concentration of aerosol particles with diameters between 0.1 and 3.0 μm and reported the change in total aerosol number concentration from upwind to downwind sites along CAO trajectories. (b-d) Scatter plots of Aitken, accumulation, and sea spray mode number concentration against precipitation accumulated within the previous 24-hours at Andenes (with values above 12 mm/hr, representing 1% of measurements, not shown). The mean number concentration in 1 mm precipitation bins is overlaid for both non-CAO (blue) and CAO (red) periods. (b-c) are plotted on the left axis scale and (d) is plotted on the right axis scale.

tion mode number, similar to the second case of Lloyd et al. (2018), which could result from higher concentrations entrained from aloft, from changes in wind direction, or differences in upwind sources.

Since the removal of aerosol particles has been linked to precipitation that occurs locally at a specific site in addition to along a calculated trajectory (Khadir et al., 2023; Tunved et al., 2012), we also examined the relationship of aerosol number

concentration to local precipitation accumulated within the previous 24 hours at Andenes during CAO events (Fig. 8b-d). Mean accumulation mode number concentration during CAO events is $117 \pm 70$ cm$^{-3}$ for accumulated precipitation above 10 mm, which is 46% lower than the mean of $216 \pm 23$ cm$^{-3}$ for accumulated precipitation below 1 mm. This decreasing trend with precipitation is consistent with the decrease in number concentration seen along trajectories, although it cannot account for potential influences from precipitation phase, intensity, duration, or spatial heterogeneity.

The mean number change along trajectories for particle sizes within the Aitken mode is relatively small ($10 \pm 26$ cm$^{-3}$), with the standard deviation indicating cases of both increases and decreases in number concentration (Fig 8a). At Andenes, mean Aitken-mode number concentration during CAO events is $211 \pm 93$ cm$^{-3}$ for accumulated precipitation below 1 mm and $43 \pm 29$ cm$^{-3}$ for accumulated precipitation above 10 mm (Fig. 8b). Prior studies have observed both an increase and decrease in the number concentration of Aitken-mode particles with precipitation accumulated along air mass trajectories (Khadir et al.,

2023; Tunved et al., 2012). During rapid boundary layer growth under CAO conditions, entrainment is also likely to play a role (Tornow et al., 2022).

For the sea spray mode, there is a small mean number increase of $1 \pm 6$ cm$^{-3}$ along CAO trajectories from Zeppelin Observatory to Andenes (Fig. 8a). At Andenes, mean sea spray mode number concentration during CAO events is $3 \pm 2$ cm$^{-3}$ for accumulated precipitation below 1 mm and $8 \pm 3$ cm$^{-3}$ for accumulated precipitation above 10 mm (Fig. 8d). Such observed

increases, both locally and along trajectories, likely result from the co-occurrence of higher winds and higher probability of precipitation, leading to the counteracting effects of sea spray particle production from over-ocean winds and their removal by precipitation. Local wind speed at Andenes is positively correlated with 24-hour accumulated precipitation (r=0.37, p<0.05; not shown) supporting the explanation that time periods with greater accumulated precipitation are associated with higher wind speeds and thus higher sea spray aerosol production.

## 6 Conclusions

The findings of this study show distinct aerosol number size distributions associated with CAO events in the Norwegian Arctic based on observations at Andenes, Norway and the Zeppelin Observatory in Svalbard for 49 CAO events that occur at Andenes and eight CAO events that have air mass trajectories between Andenes and the Zeppelin Observatory. During CAO events, aerosol number size distributions are characterized by three modes and have a statistically significant 71% higher sea spray

mode number concentration than during non-CAO conditions that is strongly correlated to the occurrence of higher over-ocean wind speeds. The mean Hoppel minimum diameter ($D_{HM}$) is 6 nm smaller during CAO events, but the higher hygroscopicity implies a lower supersaturation than during non-CAO conditions. While the integrated particle number concentration above the Hoppel minimum ($N_{HM}$) and CCN are similar between CAO events and non-CAO conditions, the ratio of $N_{HM}$/CN is approximately 15% higher with a mean of $0.61 \pm 0.25$ during CAO events due to lower CN concentrations. Along CAO

trajectories, particles in the accumulation mode experience the largest change in number concentration with a mean decrease of $93 \pm 95$ cm$^{-3}$, likely associated with precipitation scavenging.

Observations of Arctic aerosol associated with CAO events are extremely limited (e.g. Abel et al. (2017); Lloyd et al. (2018); Sanchez et al. (2022)). The analysis presented here demonstrates clear differences in the aerosol properties associated with CAO events compared to non-CAO conditions and adds eight CAO events for which upwind and downwind size-resolved

aerosol concentrations have been evaluated to the four previously reported (Abel et al., 2017; Lloyd et al., 2018; Sanchez et al., 2022). These results provide modeling studies with representative aerosol properties that are relevant for prescribing cloud droplet number concentrations in models, to which CAO simulations are particularly sensitive (de Roode et al., 2019).

Incorporating the observation-based aerosol properties of this study into future modeling work (e.g. Tornow et al. (2021)) provides an important step towards simulating CAO processes more accurately.

*Code and data availability.* COMBLE measurements are publicly available from the ARM data discovery at https://www.arm.gov/data/ (last access: February 2024) and Zeppelin Observatory measurements are publicly available from the EBAS data portal at https://ebas.nilu.no/ (last access: February 2024). Codes for the lognormal mode and hoppel minima retrievals used here are available from the UCSD digital archives (Dedrick et al., 2023, 2022b).

*Author contributions.* ASW performed the formal analysis and wrote the original draft of the manuscript. JLD, FT, IS, BS, PD, PZ, and
RK contributed to the data collection, processing, and/or analysis. LMR and AF developed the conceptualization of this work in addition to providing supervision and funding acquisition. All authors contributed to the review and editing of the manuscript.

*Competing interests.* At least one of the (co-)authors is a member of the editorial board of Atmospheric Chemistry and Physics.

*Acknowledgements.* This work was supported by Department of Energy Atmospheric System Research grant DE-SC0021983. We thank Bart Geerts and Christian Lackner for their contributions through their work on the identification of CAO events at Andenes during COMBLE.
The aerosol observations at Zeppelin station have been supported by the KAW Stiftelse (grant no. 2016.0024), the Swedish Environmental Protection agency (Naturvårdsverket), by ACTRIS-Sweden project supported by Swedish Research Council and the European Union's Horizon 2020 research and innovation program under grant agreement no. 821205 (FORCeS). We thank the staff from the Norwegian Polar Institute (NPI) for their on-site support and substantial long-term support in maintaining the measurements at Zeppelin Observatory.

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
