# Peer review of "Aerosol Size Distribution Properties Associated with Cold-Air Outbreaks in the Norwegian Arctic"

_EGUsphere, 2024_

## Author Comment (AC1)

*Reviewer comments in black, author responses in blue*

**General scientific comment:**

The manuscript presents interesting and valid results on aerosol properties measured during cold air outbreaks at the Norwegian measurement station Andenes. In addition, cases are considered when there were nearly connected flow conditions between Zeppelin Observatory and Andenes station making comparisons of aerosol properties possible concluding about the general transformation processes. Only few such studies are available making the dataset valuable especially as input data for modelling studies are scarce and needed to better represent Arctic aerosol conditions in such model simulations. I suggest minor revisions to the manuscript addressing the below mentioned comments and text corrections.

We greatly appreciate the reviewer's detailed comments on our manuscript, which caused us to strengthen our results with the addition of more clear and descriptive language, literature references, and supplemental analysis. The revisions made in response to the reviewer's comments have improved our manuscript.

**Detailed scientific comments:**

**Abstract**

Page 1, line 8:

Comment: (similar to non-CAO conditions)

Give the number for direct comparison as you have it by your statistics. You also give at other places direct comparison.

We have removed the statement "similar to non-CAO conditions" from the text and replaced it with the value for the non-CAO mean ($99\pm66$ cm$^{-3}$). The abstract has been edited to read: "…and the mean number concentration of particles that likely activated in-cloud is $109\pm61$ cm$^{-3}$ with no statistically significant difference from the non-CAO mean of $99\pm66$ cm$^{-3}$." Additionally, we've edited Section 4 to also include this value: $N_{HM}$ "…is $109\pm61$ cm$^{-3}$ during CAO events with no statistically significant difference from its mean of $99\pm66$ cm$^{-3}$ during non-CAO conditions."

**Introduction**

Page 2, line 26-27:

Comment:

Take also a look at the following papers that might be supportive:

https://acp.copernicus.org/articles/22/3067/2022/

https://iopscience.iop.org/article/10.1088/1748-9326/ac444b

We appreciate these suggestions from the reviewer. We have added a citation for both papers in our revised text.

Page 2, line 36-38:

Comment:

There is evidence that even quite small particles can serve as CCN in the Arctic as e.g. here:

https://agupubs.onlinelibrary.wiley.com/doi/10.1029/2021JD036383

https://acp.copernicus.org/articles/15/13803/2015/

Please comment!

We have now cited the suggested paper, Zabori et al. (2015), at this sentence in our text. The other suggested paper, Karlsson et al. (2022), was already referenced in this sentence where we stated that particles smaller than accumulation mode particles may also serve as CCN.

**2 Methods**

-

**2.1 CAO events**

Page 3, line 64-65:

Comment: (surface level; 2 meters above 65 mean sea level; 69.30∘N, 16.15∘E)

The closer you come to the Arctic the more uncertain trajectories will get. This is because meteorological observations needed to run the models are more scarce. This is a general problem in the Arctic. Is it reasonable to use 2m above sea level trajectories having in mind these could have touched the surface before or did you check for this bias?

We thank the reviewer for raising this point. In response to this, all trajectories have been re-run using an ensemble approach at multiple vertical levels above the surface. Previous studies (e.g. Pernov et al., 2022; https://doi.org/10.1038/s41612-022-00286-y) have initialized air mass back-trajectories at 50 m above ground level to avoid trajectories touching the surface, and we initialize our trajectories at heights up to 300 m above ground level. For our cases, even trajectories which do not ever touch the surface between Andenes and Zeppelin Observatory (since they are initialized at higher vertical levels) still pass nearby Zeppelin. Thus, we believe that our trajectories being initialized at surface level (2 m above mean sea level) does not create an issue of inaccurate trajectory paths in our study. We have added a figure to our supplemental material (Fig. S1) and added the following text in our revised manuscript to address this: "… Since a few of these 15 selected back-trajectories remain at surface level between Andenes and Zeppelin Observatory, all trajectories were initiated at multiple vertical levels in 50 m increments above the surface. For the 15 selected back-trajectories, all still pass within 200 km of Zeppelin Observatory when initiated at a height above the surface that never touches surface level at any point between Andenes and Zeppelin Observatory (Fig. S1). Moreover, the trajectories identified as not meeting the criteria of passing within 200 km of Zeppelin Observatory are unchanged for the higher initial heights."

**2.2 In-situ aerosol measurements**

Page 4, line 79-85:

Comment:

Please specify some details about inlets and drying conditions in the instruments. This might be of high relevance if you later want to make statements on changes of the PNSD in the nanometer size range, e.g. the change of Hoppel minimum, etc…. Zeppelin has ACTRIS status, but what about conditions at Andenes station? Please elaborate on this!

Details regarding the inlet and drying conditions of the Aerosol Observing System (AOS) operated by the U.S. Department of Energy (DOE) Atmospheric Radiation Measurement (ARM) Facility (deployed to Andenes for the COMBLE campaign measurements) are described in Uin et al., 2019. We have revised Section 2.2 to highlight these inlet and drying conditions and point to Uin et al., 2019 as follows: "Aerosol properties were measured at Andenes during COMBLE by the Aerosol Observing System (AOS) operated by the U.S. Department of Energy (DOE) Atmospheric Radiation Measurement (ARM) Facility. More details on the general set-up of the AOS can be found in Uin et al. (2019). … The sample flows of the UHSAS, SMPS, and CPC were dried using Nafion dryers (Uin et al., 2019), and the aerosol inlet had a particle transmission efficiency of 100% between 10 and 1000 nm (Bullard et al., 2017)."

Although the AOS does not have ACTRIS status, the conditions and setup of the AOS described in Uin et al., 2019 meets the general guidelines for aerosol inlet and drying conditions recommended by ACTRIS (guidelines accessed on 08 August 2024 at: https://actris-ecac.eu/aerosol-inlets-and-conditioning.html),

such as sampling at low relative humidity (< 40%), sampling under little diffusional and inertial losses, and excluding precipitation from the sampled aerosol by using a rain shield.

Page 4, line 92-98:
Comment:
Also, HTDMA and CCN counter are quite sensitive instruments of which performance need to be checked regularly. Provide information about calibration procedures before, during and after the measurements.

Calibrations and performance checks of the HTDMA and CCN counter during the COMBLE campaign at Andenes were handled by the U.S. Department of Energy (DOE) Atmospheric Radiation Measurement (ARM) Facility instrument mentors (Uin et al., 2019) according to the procedures described in the instrument handbooks posted on the DOE ARM website (we reference these handbooks in our manuscript as Uin, 2016b; Uin, 2016c). We have revised Section 2.2 to specifically mention this calibration information and point to the handbooks for further operational details as follows: "The HTDMA was calibrated at the beginning and end of the six-month COMBLE campaign by ARM instrument mentors, including verifying flow rates, particle sizing, and GF values of ammonium nitrate or ammonium sulfate particles against Köhler theoretical values (Uin, 2016b; Uin et al., 2019)." and "The CCN counter was calibrated at the beginning and end of the six-month COMBLE campaign by ARM instrument mentors, including applying calibration coefficients determined from counting size-selected ammonium sulfate particles before and after activation at various instrument temperatures (Uin, 2016c)."

Page 5, line 101-112:
Comment:
You calculated the sea salt mass based on the mass additions given the page before. Can you comment on how well the masses fit in terms of their abundance of number of ions? Also, you calculate here sea salt mass from the fitted mode concentrations. I guess, you transferred the number mode to a volume mode and applied a density. If so, give the details.

The reviewer is correct that we converted the number mode to a volume mode and applied a density to get modal mass. In the revised text, we explicitly state this procedure with a citation. Additionally, we have calculated the correlations between the fitted sea spray mode mass and the measured sodium and chloride ion concentrations. The following sentences have been added to Section 2.2 of the revised text: "The sea spray mode mass is calculated from the fitted sea spray mode number assuming spherical particle homogeneity and a constant sea spray density of 1.3 g cm$^{-3}$ (Dedrick et al., 2022). The Pearson correlation coefficient, r, indicates a strong correlation (r = 0.92) of the fitted sea spray mode mass with the measured sea salt mass concentration at Zeppelin Observatory (Fig. 2). There is also a strong correlation between the fitted sea spray mode mass and measured sodium ion mass concentration (r = 0.90) and measured chloride ion mass concentration (r = 0.94)." Below is a figure illustrating these additional correlations with the ion mass concentrations.

[Figure]

I would recommend to give some general quality criteria of the combination of the two particle number size distributions (submicron and supermicron) at both stations in the appendix, so the reader can evaluate how good the procedure was. You combine here particle number size distributions of two different diameters (mobility and optical). Because of all the problems that can arise, ACTRIS is not recommending to use optical particle counters at their stations for data submission. As you do so, you should convince the reader that this is a valid approach in your case.

We thank the reviewer for noting these potential issues. We have calculated the Pearson correlation coefficients (r), between measured CN concentrations and the particle number concentration integrated from the merged particle number size distributions (Merged N). At both measurement sites, there is a strong correlation between measured CN and Merged N (r = 0.80 at Andenes and r = 0.95 at Zeppelin Observatory). We have revised the text to include a discussion of these relationships and added a figure showing the correlations in the supplemental material (Fig. S2). Additionally, we have performed modal fits to only the SMPS and DMPS number size distributions and compared them with the modal fits to the merged number size distributions. For the Aitken and accumulation modes, there is very good agreement between the modal number concentration retrieved from the fits to the mobility size distributions ($N_{SMPS}$ and $N_{DMPS}$) and the merged size distributions ($N_{merged}$) at both measurement sites (see below figures). Because of the limited size range of the SMPS and DMPS, a similar comparison is not feasible for the sea spray mode. However, this mode is constrained by nephelometer measurements at Andenes and shows good correlation with sodium and chloride mass measured at Zeppelin Observatory (see Section 2.2 of our manuscript).

[Figure]

**Zeppelin Observatory**

**3 Greater sea spray particle number concentration**

Page 6, line 140:

Comment:

Please check, I feel you exchanged marine and continental wind directions at least in the text. If this is the case, please recheck your data analysis for consistency.

We thank the reviewer for their careful considerations. However, we have rechecked this and believe that we did not exchange marine and continental wind directions. Our statements in the text and our data analysis are consistent. We define $0°/360°$ as North, and so $250°$-$30°$ is from the northwest/north which is towards the open ocean from Andenes (which we term as "marine" directions) and $100°$-$230°$ is to the southeast/south which is towards Scandinavia from Andenes (which we term as "continental" directions). Continental wind directions have lower sea spray mode number concentration than do marine wind directions, which is already correctly stated in the text. To improve clarity, we have added "northwesterly" and "southeasterly" to the revised text as follows: "Binning the sea spray mode number concentration by wind direction ($w_{dir}$) as either marine ($250°>w_{dir}>30°$, northwesterly) or continental ($100°>w_{dir}>230°$, southeasterly) reveals…"

General comment:

The information here on wind directions for CAO events and non-CAO conditions, numbers observed in different modes, significance between variables, etc. could be put in a table for better overview.

We have incorporated this suggestion into the revised manuscript (see Table 1).

**4 Activation of particle to cloud droplets**

Page 8, line 159 - 160:

Comment:

I do not get how you derive the Hoppel minimum diameter. You have to elaborate on this. It seems obvious from Figure 5 that it is somewhat smaller for CAO events compared to non-CAO conditions, but you need to explain how you quantitatively come up with a value. Are there some references about the procedure you use?

We appreciate the reviewer noting that our description was unclear and have added a more detailed description as well as references for the methods we used in Section 2.2 of the revised manuscript as follows: "The Hoppel minimum diameter ($D_{HM}$; local minimum in concentration located between the peaks of the Aitken and accumulation modes, Hoppel et al. (1986)) is retrieved for all measured aerosol particle number size distributions at Andenes following the methods presented by Dedrick et al. (2024). Using the $D_g$ of the fitted Aitken and accumulation modes as upper and lower diameter bounds, $D_{HM}$ is taken as the diameter bin of the merged measured number size distribution that has the lowest particle number concentration within these diameter bounds." Additionally, the code used for the Hoppel minimum retrieval calculations is publicly posted on the UCSD digital archives. We mention this along with a citation at the end of the manuscript under the *Code and data availability* statement.

Page 9, line 174 - 175:
Comment:
How sensitive is effective SS against uncertainties in kappa derived from HTDMA and Hoppel minimum diameter. Also, if you calculate kappa from HTDMA and CCN counter for the same aerosol type, you will receive different results. Put this in perspective to your results and their uncertainties! I request here some kind of sensitivity study. Here are some papers on kappa discrepancies using the two different techniques:
https://acp.copernicus.org/articles/23/4931/2023/
https://pubs.rsc.org/en/content/articlelanding/2020/em/d0em00179a
We thank the reviewer for asking us to add more details about uncertainties in our results. In the revised manuscript, we use a Monte Carlo approach to estimate the uncertainty in effective supersaturation similar to the methods of Massling et al. (2023) and Schulze et al. (2020). We find an uncertainty of less than ±15%, given the uncertainties in $D_{HM}$ (±2.5%) and κ (±24%). Please see the revised Section 2.2 for more discussion on this. Additionally, we have now pointed out that discrepancies in κ calculated from HTDMA vs CCN measurements may impact our results on the effective supersaturation differences between CAO events and non-CAO conditions. We have added the following text to our supplemental material and reference it in Section 4 of the revised main text:
"The lower effective supersaturation during CAO events is based on HTDMA-derived κ values and that previous studies have found significant discrepancies in κ values calculated from HTDMA versus CCN measurements (e.g. Petters and Kreidenweis, 2007; Massling et al., 2023). CCN-derived κ values are not available for this study due to limited CCN measurements, however, if they varied significantly from the HTDMA-derived κ values of this study, there is potential for the estimated effective supersaturation during CAO events to be higher than that during non-CAO conditions."

Page 9, line 177-179:
Comment: Please check this sentence, I think you make the wrong conclusion based on theory.
Thank you for pointing this out. The "lower kappa" statement was a typo, as it should have instead been "higher kappa". We have now corrected this to read "the *higher* kappa during CAO events would imply a lower effective supersaturation", which agrees with theory.

Page 9, line 179 – 180:
Comment:
I do not think you can say this. This is dependent on the quantitative change of the parameter and here you compare apples and pears!
We agree with the reviewer that the change in supersaturation is dependent on the quantitative change of Kappa and $D_{HM}$, which have different unit scales and will require different magnitudes of change in each

to produce the same change in supersaturation. We understand that our original statement may have been vague, thus inadvertently creating an overgeneralization. In the revised text we emphasize that of the observed increase in hygroscopicity and observed decrease in $D_{HM}$, specifically for the range of values associated with the CAOs of our study, the increase in hygroscopicity has the stronger effect on supersaturation, which agrees with the observed lower supersaturations associated with CAO events. The edited text now reads as follows: "If the aerosol chemical composition and by consequence its hygroscopicity, represented by $\kappa$, were the same between CAO events and non-CAO conditions, then the smaller $D_{HM}$ during CAO events would imply a higher effective supersaturation during CAO events (Fig. 6). However, if $D_{HM}$ were the same between CAO events and non-CAO conditions, then the higher $\kappa$ during CAO events would imply a lower effective supersaturation during CAO events (Fig. 6). There is a statistically-significant lower median effective supersaturation of 0.37% during CAO events compared to a median of 0.41% during non-CAO conditions (Fig. 5). Interestingly, this suggests that for the range of $D_{HM}$ and $\kappa$ observed during the CAO events, the higher hygroscopicity ($\kappa$) has a larger effect on supersaturation than does the smaller particle size ($D_{HM}$), resulting in the observed lower effective supersaturation during CAO events than during non-CAO conditions."

Page 10, line 188 – 189:
Comment:
Please see also the following paper where the CCN counts are listed depending on season:
https://acp.copernicus.org/articles/23/4931/2023/
Thank you for bringing this paper to our attention. We have added it to our table of reported CCN values (Table S2) and now cite it in the text.

Page 10, line 193 – 195:
Comment:
I do not see the difference in this ratio as a consequence of lower CN concentrations during CAO events in general. Can you explain why that is? If the pure fact that there are less particles in this air mass has something to do with the fraction activated, then please give the scientific reasons for this.
The difference in the ratio $N_{HM}$/CN is a consequence of the lower CN concentrations during CAO events because the numerator of the fraction, $N_{HM}$ remains similar during both CAO and non-CAO conditions. If the numerator remains similar but the denominator becomes smaller, then the result is that the fraction is larger (as is observed). We understand that this may not have been clear in the text since we state that $N_{HM}$ is similar in value several sentences before discussing the fraction. To improve clarity, we have edited the text to remind the reader that $N_{HM}$ (the numerator of the fraction) is similar between CAO and non-CAO conditions when discussing the difference in the CN concentration and $N_{HM}$/CN ratio. This sentence now reads: "As a consequence of similar $N_{HM}$ concentrations between CAO events and non-CAO conditions and lower CN concentrations during CAO events, the mean ratio…".

**5 Scavenging by precipitation**
Page 11, line 203 - 204:
Comment:
I feel this is not fully correct. You can only give a decrease in total particle number for a given size range or a mode. When you refer to a specific size (in this case 142 nm) you can only talk about a change in dN/dlogDp. Also in Figure 8a, you label the y axis as delta N, should it not be delta dN/dlogDp for the size distribution, and similarly it needs to be delta N only for the number concentration observed in modes.

Thank you for pointing this out, we agree with the reviewer that in this sentence and in Figure 8a we did not make a clear distinction between N vs dN/dlogDp. We have updated Figure 8a to have two axes, a left axis (delta dN/dlogDp) for the change in the aerosol number size distribution and a right axis (delta N) for the change in total aerosol concentration for the modes and literature values. The figure caption has been edited accordingly to reflect these changes. Additionally, we have edited this paragraph of the text in the revised manuscript to make a clear distinction between changes in the aerosol number size distribution (dN/dlogDp) and changes in the total aerosol number concentration of the accumulation mode (N).

Page 12, Figure 8a:
Comment:
It seems that other studies have not done that analysis with a similar size resolution compared to this study. Please add their approach shortly in the Figure caption.
This is true. We have now added a brief description of the approach of Abel et al. (2017), Lloyd et al. (2018), and Sanchez et al. (2018) to the caption of Figure 8 as follows: "Overlaid in black lines with markers are the mean change in total aerosol number concentration reported for the case of Abel et al. (2017), cases two and three of Lloyd et al. (2018) and the closed- to open-cell case of Sanchez et al. (2022). These studies only measured the total number concentration of aerosol particles with diameters between 0.1 and 3.0 µm and reported the change in total aerosol number concentration from upwind to downwind sites along CAO trajectories."

Page 12, Figure 8bcd:
Comment:
Are you sure the means are right? Please recheck. In Figure 8b there seems to be an artifact around 100 #/ccm, similar to Figure 8c at 200 #/ccm.
We thank the reviewer for their careful considerations. We have rechecked these means, and they are correct. The means in the first 1mm precipitation bin are affected by a relatively larger range in number concentration and the density or distribution of the points within that range. In the first 1mm bin, the blue non-CAO mean is lower due to higher density of points towards lower number concentration values. In contrast, the red CAO mean is higher due to fewer points that are distributed more evenly between 0 and 500 $cm^{-3}$. In reviewing Figure 8, however, we did notice an error such that the y-axis limits on Figure 8d did not show points below 5 $cm^{-3}$. We have fixed these y-axis limits to start at 0 $cm^{-3}$ in the revised manuscript so that all data is visible in this figure.

**6 Conclusions**
Page 13, line 239-240:
Comment: Was it 15 air mass trajectories or better eight events with connected flow conditions?
While both are correct (we analyzed 15 air mass trajectories that occurred during eight CAO events), we agree with the reviewer that in this sentence it is better to mention the eight CAO events. We have edited this sentence to the following: "… based on observations at Andenes, Norway and the Zeppelin Observatory in Svalbard for 49 CAO events that occur at Andenes and eight CAO events that have air mass trajectories between Andenes and the Zeppelin Observatory."

Table S2:
Comment: How can Paramanov measure up to 80% more CCN compared to CN?
Thank you for pointing this out, the decimal point was incorrectly placed. This typo has now been corrected to ~0.08-0.18.

**General language comment:**

I prefer always a detailed description/wording of aerosol size distributions. You can look at number, surface, volume, mass, etc.…. I made a number of text suggestions below as you talk about particle number size distributions here, but then also say so.

We appreciate these specific language comments from the reviewer to improve the clarity of our text. We have incorporated each of the below comments into our revised manuscript.

**Specific language comments:**

**Abstract**

Page 1, line 1:

… aerosol particles serving as …

Page 1, line 3:

… aerosol number size distribution modes …

Page 1, line 4:

… upwind from Andenes in …

Page 1, line 11:

… aerosol number size distribution …

**Introduction**

Page 2, line 34:

… aerosol number size distribution …

Page 2, line 46:

… aerosol number size distribution …

Page 2, line 51:

… aerosol number size distribution …

Page 2, line 54:

… droplets …

**2 Methods**

-

**2.1 CAO events**

Page 4, line 69:

… aerosol number size distribution …

**2.2 In-situ aerosol measurements**

Page 4, line 74:

Particle number size distribution …

Page 4, line 76/77:

… to size and count …

Page 4, line 79:

… particle number size distributions …

Page 4, line 99:

Particle number size distribution …

**2.3 Ancillary variables**

-

**3 Greater sea spray particle number concentration**

Page 6, line 127:

… aerosol particle number size distribution …

Page 7, Figure 3:

Median particle number size distribution ...

Page 7, line 150:

… at Andenes …

**4 Activation of particle to cloud droplets**
Page 7, line 155:
… aerosol particle number size distribution …
Page 8, line 162:
… aerosol number size distributions …
Page 8, line 164:
… or aerosol microphysical properties …
Page 9, Figure 5:
… lower and upper edges …
Page 10, line 195:
… are activated to cloud droplets …

**5  Scavenging by precipitation**
Page 11, line 200:
… particle number size distributions …
Page 11, line 201:
… passed …
Page 11, line 207:
… in accumulation mode particles during our …
Page 11, line 217:
… of trajectory arrival at Andenes …
Page 12, line 229:
… from Zeppelin Observatory to Andenes …

**6 Conclusions**
Page 13, line 238:
… aerosol number size distributions …
Page 13, line 240:
… aerosol number size distributions …
Page 13, line 241:
… compared to non-CAO events …
Figure caption S1:
… aerosol number size distribution …

---

## Author Comment (AC2)

*Reviewer comments in black, author responses in blue*

Congratulation to the authors, a good contribution to the field, adding important Cold air outbreaks CAO information. Particularly appreciated is the comparison among two monitoring sites and the good number (49) of event reported. Overall, it is a good contribution to the field following the important papers well cited of Abel 2017, Sanchez 2022 and Llloyd 2018.

We thank the reviewer for their insightful comments. We have revised our manuscript accordingly (see below) and believe that the manuscript has benefited from the addition of background information, discussion of literature, and references in response to the reviewer's specific comments.

Two main small comments:

1. Please expand the introduction and explain better what the COA events are and why they are important, perhaps a conceptual diagram or a figure explaining better the event, poorly explained in the current literature.

We have revised the first paragraph of our introduction to include a more detailed description of CAO events and their relevance and importance. Additionally, we have included more literature references and included a citation directly to Figure 1 of Geerts et al. (2022) which illustrates through satellite imagery one of the CAO events we analyze in our study. The revised paragraph now reads:

" The equatorward transport of polar air masses brings cold air over a relatively warmer open ocean, spurring large turbulent surface fluxes of heat and moisture and creating so-called marine cold-air outbreak (CAO) events. The forming clouds often appear first in the shape of rolls (Brümmer, 1999) before filling in towards a near-overcast deck and then transitioning into a more broken, open-cellular state further downwind (Abel et al., 2017; Lloyd et al., 2018; McCoy et al., 2017; Geerts et al., 2022). Such a cloud regime transition during CAO events is often seen in satellite imagery (e.g. Fig. 1 of Geerts et al. (2022)) and has important implications for the regional radiation budget (McCoy et al., 2017). Climate and numerical weather prediction models struggle to accurately capture the clouds associated with CAO events. Such model limitations emerge since important dynamical aspects are unresolved (Field et al., 2017). In addition, there is evidence of strong cloud and aerosol microphysical processes that may modulate the macrophysical structure of clouds. For example, Tornow et al. (2021) found that mid-latitude CAO cloud transitions which resembled observations could only be simulated when the number concentration of aerosol serving as cloud condensation nuclei (CCN) was allowed to evolve with time, as aerosol can control the onset of transition-driving precipitation (Abel et al., 2017; Tornow et al., 2021). For polar CAOs of lower temperature, aerosol may additionally provide a reservoir for ice nuclei (IN; e.g. Bigg (1996)). Previous modeling of cold-air outbreaks have typically relied on sparse observational constraints (e.g. Field et al. (2014); Abel et al. (2017); Tornow et al. (2023)), illustrating a need for observations of aerosol properties associated with CAO events in order to inform, constrain, and improve model simulations and forecasts."

2 I was surprised by the sea spray mode at 450nm, sensibly larger to what I understand being the "referement" mode at 160nm (Prather et al 2013, etc). Perhaps a discussion on this and an appropriate small summary of the art on the typical sea spray aerosol mode may be necessary.

Thank you for pointing this out. We have added the following sentence to Section 2.2 of the revised manuscript: "This sea spray mode centered at diameters between 0.3 and 0.8 µm is consistent with

previous reports of sea spray mode mean diameters ranging between 0.14 and 0.6 μm in field measurements (Russell et al., 2023; Modini et al., 2015; Dedrick et al., 2022)." Our median sea spray mode during CAO events centered at 450 nm is consistent with previously observed sea spray mode mean diameters in field measurements. The sea spray mode mean diameter is found to be 300 nm in Lewis and Schwartz (2004), 300 nm in Quinn et al. (2017), 500 nm in Saliba et al. (2019), 600 nm in Sanchez et al (2021), and 400 nm in Dedrick et al. (2022). See Table 3 of Russell et al. 2023 for further details (https://doi.org/10.1016/j.earscirev.2023.104364). Prather et al. 2013 reports a mean sea spray mode at 160 nm, but it is worth noting that it is based on laboratory wave simulations rather than field measurements. In agreement with this, Salter et al. (2015; https://doi.org/10.5194/acp-15-11047-2015) also report a smaller mean sea spray mode centered at approximately 100 nm when using a sea spray simulation chamber.

Congratulation, it is a well described paper with a very good dataset collected at two stations. We appreciate the reviewer taking the time to review our manuscript.